# Bionic Birdlike Imaging Using a Multi-Hyperuniform LED Array

**DOI:** 10.3390/s21124084

**Published:** 2021-06-14

**Authors:** Xin-Yu Zhao, Li-Jing Li, Lei Cao, Ming-Jie Sun

**Affiliations:** School of Instrumentation and Optoelectronic Engineering, Beihang University, Beijing 100191, China; zhaoxinyu@buaa.edu.cn (X.-Y.Z.); lilijing@buaa.edu.cn (L.-J.L.); caolei_bh17@163.com (L.C.)

**Keywords:** multi-hyperuniform, single-pixel imaging, frequency aliasing, color misregistration

## Abstract

Digital cameras obtain color information of the scene using a chromatic filter, usually a Bayer filter, overlaid on a pixelated detector. However, the periodic arrangement of both the filter array and the detector array introduces frequency aliasing in sampling and color misregistration during demosaicking process which causes degradation of image quality. Inspired by the biological structure of the avian retinas, we developed a chromatic LED array which has a geometric arrangement of multi-hyperuniformity, which exhibits an irregularity on small-length scales but a quasi-uniformity on large scales, to suppress frequency aliasing and color misregistration in full color image retrieval. Experiments were performed with a single-pixel imaging system using the multi-hyperuniform chromatic LED array to provide structured illumination, and 208 fps frame rate was achieved at 32 × 32 pixel resolution. By comparing the experimental results with the images captured with a conventional digital camera, it has been demonstrated that the proposed imaging system forms images with less chromatic moiré patterns and color misregistration artifacts. The concept proposed verified here could provide insights for the design and the manufacturing of future bionic imaging sensors.

## 1. Introduction

Both spatial and spectral information provides us crucial knowledge of the world. Chromatic digital cameras obtain spatial and spectral information simultaneously by placing an absorbing color-filter array (a.k.a. the Bayer filter) on top of a detector array [1], because photoelectronic sensors are only sensitive to light intensity, regardless of its wavelength. Due to technical limitations and commercial considerations, the elements of the pixelated detectors, as well as those of the filter arrays, are usually arranged in a Cartesian geometry. Such periodic arrangement of both the detector array and the Bayer filter array introduces artificial effects decreasing the fidelity of the captured images. Frequency aliasing, which converts frequencies above the Nyquist limit into moiré fringes during the optical sampling, is one such effect. Color misregistration, which causes inauthentic color shifts during the demosaicking process, is another. Many post-processing algorithms have been proposed [2,3,4,5,6,7,8,9] to suppress these artificial effects in the captured images, which increase the computational burden on imaging system. However, these effects could be avoided or suppressed if raw image data could be sampled in a different manner with low cost.

The evolution of species is governed by neither technical limitation nor commercial consideration, but environmental requirements, and environment requires diurnal animals to evolve eyes which can obtain images with high fidelity and dynamic performance. Some researches about human retinas showed that arrangement of the photoreceptors is random and uniform, which is able to yield images with better reconstruction quality by suppressing frequency aliasing [10]. The similar structures exist in the avian vision system as well. Recent biological investigations have found that birds, being the vertebrate with the most sophisticated vision, have retinas consisting of five types of cones, each of which independently exhibits a disorder on small-length scales but a quasi-uniformity on large scales [11,12]. The fantastic structure of correlated disorder is known as hyperuniformity [13], and the fact that such arrangement can obtain high fidelity images is explained by the theory that a slight irregularity in the optical sampling arrangement can avoid frequency aliasing [14]. It has been shown that the evolution of the avian vision system may be most sophisticated among all animals. Another biological research [15] shows that the birds can achieve continuous imaging with up to 145 fps frame rate. A recent hyperuniform sampling experiment [16] further verified the feasibility of the theory. However, the experiment was performed with a single-pixel imaging [17,18,19,20,21] system using a digital-micromirror-device which is commonly used in many applications [22,23]. While the sampling patterns were designed to be hyperuniform, the micromirrors forming the patterns were arranged in Cartesian coordinates and displayed at an up-to-22 KHz modulational rate. Consequently, the frequency aliasing in the sampled image was not suppressed completely and the system cannot perform high dynamic tasks.

In this paper, we addressed that the periodic sampling caused frequency aliasing and color misregistration by utilizing a customed chromatic LED array in which red-green-blue luminous points formed a multi-hyperuniform arrangement [11], that is, luminous points of one color form a hyperuniform point pattern, and all points together, regardless of their colors, exhibit hyperuniformity as well. Such arrangement of the chromatic LED array was designed to mimic the multi-hyperuniform structure of the chicken retina system [12]. Optical sampling using multi-hyperuniformity was performed experimentally via a single-pixel imaging system. The high-speed hyperuniform LED array developed in this work, which have a maximum illumination rate of 2.5 MHz, can effectively improve the dynamic performance of the imaging system. Both numerical and experimental results indicated that the images retrieved by multi-hyperuniform sampling contained less chromatic moiré patterns at high frequencies and less color misregistration artifacts at the edge of color transition, where the proposed imaging system achieved 208 fps frame rate in experiment. The work is different from the methods of optimizing interpolation algorithm, which can solve these artificial effects in hardware through a simple imaging system. The proof-of-principle system demonstrated here might push us one step closer to the biomimetic digital camera which the imaging community aimed to invent for so long.

## 2. Theory

Hyperuniform structure exists in not only avian retinas but also physical systems such as crystal [24], or even the large-scale structure of the universe [13]. The property of a hyperuniform system can be quantified as the density fluctuation in its corresponding point patterns. For a 2D point pattern with hyperuniform distribution, the variance of the number of points *σ*^2^*(R)* within a circular domain *S* is approximately proportional to the *R* [13] i.e.,
(1)$$\sigma^{2}\left(R\right)=\langle N_{S}^{2}\rangle -{\langle N_{S}\rangle }^{2}\propto R$$
where *N_s_* is the number of points contained in *S*, and angular brackets represent an ensemble average. *R* is the radius of a circular observation window. Equation (1) indicates that the variance of the hyperuniform point patterns grows more slowly than the area of the domain, while for any statistically homogeneous and isotropic point pattern, the variance cannot grow more slowly than the area of circle *S* or other strictly convex domains [25,26].

A special hyperuniformity, known as multi-hyperuniformity [11], contains more than one type of points. For example, five types of cone photoreceptors exist in chicken retina: violet, blue, green, red species to sense color and double species to detect luminance. The point patterns of these five types of cones are arranged individually and never occurred in the near vicinity of other cones of the same type, which ensure each cone pattern achieves a much more uniform arrangement [11]. All types of photoreceptors grow simultaneously with the constraints of cell size, and such competing interactions ensure that all cone patterns are arranged in a hyperuniformity. Multi-hyperuniform structure contains multiple point species where both the total population and the individual point types are simultaneously hyperuniform. It is worth noting that the overall point arrangement in multi-hyperuniform obey Equation (1) as well, which means that the point patterns are independent with each other and display hyperuniform in total whether the individual species is removed or not. Such multi-hyperuniform structure is believed to be the main reason that birds have the most sophisticated vision of any vertebrate. It would be interesting to exploit the concept in optical sampling by using such geometry, and here it was performed in the manner of structured illumination in a single-pixel imaging scheme.

A chromatic LED array was developed, in which its red-green-blue luminous points were arranged in multi-hyperuniformity. Specifically, a hyperuniform point pattern was generated by a ‘cell-growing’ random procedure based on a regular hexagonal geometry [16]; the green luminous points, being the center of the LED chips, were then arranged following the hyperuniform point pattern, as shown in Figure 1a. As the red-green-blue luminous points in an LED chip have a fixed geometry (Figure 1b), a periodicity would exist in the LED array if each chip were arranged in the same manner. Therefore, an extra irregularity was introduced by rotating the LED chip of the *i*th column and the *j*th row with a randomly generated angle *θ_ij_* (Figure 1c). The LED array, as shown in Figure 1d, was fabricated by placing 32 × 32 LED chips on corresponding positions and integrating them on the printed circuit board, where the red-green-blue luminous points on each chip have center wavelengths of 632 nm, 518 nm, and 468 nm, respectively. It is worth mentioning that, for the purpose of variance *σ*^2^*(R)* estimation and further numerical simulation, the multi-hyperuniform arrangement were generated on an underlying Cartesian grid of 544 × 544 pixels with one pixel corresponding to 0.1 mm × 0.1 mm. The actual LED array, however, was not limited by Cartesian coordinates. Recent research has found that the quantities of five types of cones are different in avian retina, where the double cones were the most abundant cone type (40.7%) followed by green (21.1%), red (17.1%), blue (12.6%) and violet (8.5%) single cones. Due to the fixed arrangement of R-G-B channels of each LED chip, the three channels have the same spatial density on the multi-hyperuniform LED array with 32 × 32 chips.

To evaluate the hyperuniformity of the chromatic LED array, as in the calculating process in [11], the variances *σ*^2^*(R)* were computed directly for each monochromatic point patterns separately, and for overall point pattern, as shown in Figure 2. Specifically, for each *R* value, 2500 circular domains *S* were randomly placed in the pattern without overlapping the system boundary. The maximum radius was chosen to be *R_max_* = *L*/2, limited by the pattern size *L*. The variances of each pattern were fitted (dashed lines in Figure 2) using the fitting function:(2)$$\sigma^{2}\left(\frac{R}{D}\right)=P\left(\frac{R}{D}\right)\left(1+Q\cos\left(\frac{\pi }{2}\frac{R}{D}+\frac{\pi }{3}\right)\right)$$
where a cosine term represented that the patterns were originated from the regular hexagonal arrangement [16], the window size *R* is normalized by the averaged points’ distance *D* = 17 for monochromatic point patterns and *D* = 5.67 for overall point pattern. According to the previous research [11], the structural properties of individual and overall point patterns could be obtained in the same manner, so the Equation (2) is suitable for hyperuniform evaluation of chromatic LED point array.

The parameters of the fitting curves in Figure 2 are listed in Table 1. The fitting curves and their fitting parameters, listed in Table 1, indicated that the variances of all four patterns grew proportionally to *R* rather than *R*^2^, which met the criterion of hyperuniformity described by Equation (1). Each monochromatic luminous point pattern exhibits hyperuniformity individually, and combined as an overall point pattern, the LED array remains hyperuniform, therefore, the arrangement of the LED array was multi-hyperuniform.

It is worth mentioning that *Q* is the coefficient for the cosine term in Equation (2), representing the hexagonal geometry. In Table 1, coefficient *Q* of the green point pattern is larger than the red and blue ones because the latter ones had an extra random rotation introduced during the generation of the pattern, meaning the red and blue point patterns had a larger deviation from the regular hexagonal geometry than the green point pattern. The parameter *P* is a coefficient for fitting curves.

## 3. Numerical Simulations

Numerical simulations were performed using a multi-hyperuniform sampling point pattern, which was generated based on the multi-hyperuniform LED array. The point pattern, having a 544 × 544 underlying pixel grid, contained 3072 sampling points, 1024 for red, green, and blue each. Figure 3a showed a partial area of the arrangement with 4 red, 4 green, and 4 blue points. Each LED luminous point, not strictly a point, had the size of 6 pixel grids, representing its 0.3 mm × 0.2 mm physical size.

For comparison, two other types of sampling patterns were also used in the numerical simulation. The regular LED pattern, a part of which was shown in Figure 3b, represented that the LED chips were arranged in a regular geometry. The regular LED pattern also contained 3072 sampling points, 1024 for red, green, and blue each. The Bayer pattern, a part of which was shown in Figure 3c, is a common arrangement used in conventional chromatic digital cameras. The Bayer pattern, having a 1:2:1 ratio of red, green, and blue points, contained 3072 sampling points which consisted of 768 red, 1536 green, and 768 blue points.

To ensure that the comparison is fair, the three sampling patterns had the same number of sampling points and the same size of underlying pixel grid, therefore, the same sampling frequency and the same field-of-view for the optical sampling.

In numerical simulation, a group of 35 chromatic images, whose pixel resolution is 544 × 544, were used as the objects. Each chromatic image *I* was under-sampled by the three sampling patterns, and a demosaicking algorithm [4,5] was applied to the under-sampled monochromatic data of red, green, and blue to reconstruct the chromatic image *I’*. There are many sophisticated demosaicking methods for various applications [6,7,8] and the gradient-based interpolation algorithm [9] was applied in this work. The gradients of different directions are calculated according to the sampling structure, which can ensure for selecting the proper direction to estimate the missing pixel values of the images. This algorithm is efficient to reduce the pseudo color of color-transition area in reconstructed images for three sampling patterns without much time cost. It is worth mentioning that the demosaicking algorithm used here is not best for three sampling structures but valid and simple, which can ensure the fair comparison of image reconstruction with three sampling patterns. There are some complicated algorithms to improve the quality of images as well, which would cause more time costs of imaging. It is verified that the multi-hyperuniform sampling structure could be used to improve the image quality in a hardware way without more computational burden.

The qualities of reconstruction images were evaluated using the root mean square error (RMSE) between each original image *I* and its reconstruction *I’* as:(3)$$RMSE_{channel}=\sqrt{\frac{{\sum_{i,j=1}^{m,n}\left({I^{\prime }}_{channel}(i,j)-I_{channel}(i,j)\right)}^{2}}{m\times n}}$$
where m=n=544 were the pixel resolution of the images, and channel, being red, green, or blue, represented the monochromatic data of different channels. The final RMSE is the averaging value of the RMSEs for three monochromatic channels, i.e.,
(4)$$RMSE=\sum_{channel}^{R,G,B}RMSE_{channel}/3$$

Both original images and the reconstructions were normalized to the same scale.

The RMSEs of all resulting images, sorted in descending order of multi-hyperuniform pattern RMSEs, are shown in Figure 4. In most cases, 33 out of 35, the proposed multi-hyperuniform pattern yielded the best resulting images among the three sampling patterns. The average RMSE for all 35 images reconstructed from multi-hyperuniform sampling, being 0.1214, is the smallest of the RMSEs yielded from the three sampling patterns. Regular and Bayer patterns each yielded the lowest RMSE in two cases, where the original images contain large blocky areas with no color transitions and high-frequency details. The slight degradation of image quality by using multi-hyperuniform sampling in such cases, was predicted by the fact that the irregularity in optical sampling degrades image quality [27]. The RMSEs of the reconstructed images by multi-hyperuniform structure demonstrate the improvement of the image quality on a pixel-wise level.

For considering larger-scale features of reconstructed images, the structural similarity (SSIM) between each original image *I* and its reconstruction *I’* is calculated as:(5)$$SSIM({I^{\prime }}_{channel},I_{channel})=\frac{\left(2\mu_{I_{channel}}\mu_{{I^{\prime }}_{channel}}+c_{1}\right)\left(2\sigma_{I_{channel},{I^{\prime }}_{channel}}+c_{2}\right)}{\left(\mu_{I_{channel}}^{2}+\mu_{{I^{\prime }}_{channel}}+c_{1}\right)\left(\sigma_{I_{channel}}^{2}+\sigma_{{I^{\prime }}_{channel}}^{2}+c_{2}\right)}$$
where, μ is the average value of images, $\sigma^{2}$ is the variance of images, $c_{1}$ and $c_{2}$ are constants. $\sigma_{I,I^{\prime }}$ is covariance of *I* and *I’*, $c_{1}=0.01$ and $c_{2}=0.03$ are constants for ensuring the validation of this equation. The meaning of the channel is the same as the previous calculation process. The final SSIM is the averaging value of the SSIMs for three channels, i.e.,
(6)$$SSIM=\sum_{channel}^{R,G,B}SSIM({I^{\prime }}_{channel},I_{channel})/3$$

The SSIMs of all images reconstructed are shown in Figure 5, where image indexes are consistent with the results in Figure 4. The averaging SSIM for all 35 images from multi-hyperuniform sampling is the largest compared with other results yielded from regular and Bayer sampling. The two images with the large block can be reconstructed better by regular sampling pattern and their indexes are the same with the two cases where we find lower RMSEs by using regular and Bayer patterns in Figure 4. The SSIMs of the reconstructed images demonstrate that image quality can be improved on larger-scale features by multi-hyperuniform sampling as well.

Since aliasing errors are usually caused by insufficient sampling, it is necessary to compare the proposed sampling with other random sampling strategies. The blue-noise sampling strategy is another random strategy, which is derived by the human vision system. This method is used here and the reconstruction results are listed in Figure 6 and Figure 7. The points in blue-noise sampling pattern are arranged randomly but the distances of any two adjacent points are uniform, which causes the reconstructed images to be close to the results using multi-hyperuniform sampling structure.

Due to the fact that different images have different power distribution of image spatial frequency [28,29], more various scenes are used for comparisons, where the reconstructed images of artificial scenes and natural scenes are obtained by different patterns in Figure 6 and Figure 7. In Figure 6, it is obvious that less color misregistration artifacts and moiré fringes were observed in reconstructed images by multi-hyperuniform sampling pattern compared with the other structures. It is worth noting that the color misregistration could be considered as one of the small-scale features and the chromatic moiré fringes could be treated as one of the large-scale features. The RMSEs and SSIMs are sequentially listed under the resulting images, which indicate that multi-hyperuniform sampling structure can suppress the color misregistration and chromatic moiré fringes in different scale features. The reconstructed results in Figure 7 demonstrated that the multi-hyperuniform sampling structure is valid to improve the qualities of natural scenes with sparse periodic patterns as well.

## 4. Experimental Results

It is difficult to perform the multi-hyperuniform optical sampling with an actual detector array, because such a chromatic detector array would be impossible to manufacture. However, it is possible to perform a multi-hyperuniform optical sampling experiment utilizing the computational imaging methods which have emerged during the last decade [30,31,32,33]. Single-pixel imaging, being a typical computational imaging method, forms an image by sampling the scene with varying structured illuminations, and associating the illumination patterns with the corresponding light intensities recorded with a single-pixel detector. This imaging strategy provides advantages for imaging in situations that are challenging with a detector array, such as special spectrum imaging [34,35], adaptive imaging [36,37,38], optical phased array imaging [39] and 3D profiling [40,41,42,43].

In this work, a single-pixel imaging experimental system was set up as shown in Figure 8. The multi-hyperuniform chromatic LED structured illumination module was used in the system to sample the scene, which consists of a field-programmable gate array (Xilinx Spartan XC6SLX9-2FTG256C), a drive circuit and an LED array developed in Section 2. During the experiment, for each monochromatic channel, a projection lens (f = 150 mm) projected *N* masks *P_i_* (I = 1, …, N) displayed on the LED array. The mask *P_i_* is orthonormal and derived from the Hadamard matrix, a square matrix with elements ±1 whose rows (or columns) are orthogonal to one another [44]. The LED array displays monochromatic illumination masks at the rate of 1.25 MHz. The high illumination rate was achieved by using the line control modulational strategy proposed in our previous work [45]. A single-pixel bucket detector (Thorlabs PMT2102) and a digitizer (PicoScope 6404D) were used to record the corresponding reflecting total light intensities *S_i_* (I = 1, …, N). The image *I’_channel_* for the monochromatic channel can be reconstructed as:(7)$${I^{\prime }}_{channel}=\sum_{i=1}^{N}S_{i}\cdot P_{i}$$
where the *N* is the quantity of the Hadamard basis masks. The problem of reconstructing the single channel image of the scene becomes a problem of solving *N* independent unknowns using a set of linear equations. Due to the orthonormal property of the mask, the Equation (7) can be solved perfectly if the number of Hadamard basis masks is equal to pixels of the image [16,21,44,45]. Provided the scene is sparse, compressive sensing [18,21,44] can be used to reconstruct image with *N* < 32 × 32 measurements by sub-sampling the scene and solving the optimization problem. To yield one 32 × 32 monochromatic image, 2048 masks (1024 Hadamard masks and their inverses) were used to perform a fully Hadamard sampling. After images *I’_channel_* for red, green, and blue were reconstructed separately, the gradient-based interpolation algorithm was applied to yield the chromatic image *I’* of the object. A chromatic image required 6144 illumination masks, or 4.8 ms acquisition time, resulting in a 208 fps frame rate for the multi-hyperuniform chromatic LED array-based single-pixel imaging system. Due to the lab manufacturing limitation, the imaging experiment only can achieve 32 × 32 pixels resolution.

For comparison, another LED array with regular sampling arrangement, as shown in Figure 3b, was used in the experimental system to obtain images with regular sampling. An ordinary smartphone camera was also used to capture images with Bayer sampling, where the method of sub-sampling is chosen to ensure the same sampling structure compared with the simulation. All images were obtained with the same number of spatial sampling points to make sure the comparison was fair.

Figure 9 illustrated the resulting images yielded from multi-hyperuniform, regular and Bayer sampling patterns. Like numerical simulation, color misregistration and chromatic moiré fringes were observed in images yielded by regular and Bayer sampling, while multi-hyperuniform sampling suppressed both artifacts. The calculated RMSEs and SSIMs listed below the images are in a good agreement with the numerical simulation. It is showed below that the images reconstructed by multi-hyperuniform have the lowest RMSE and highest SSIM values, demonstrating the effectiveness of suppression of color misregistration and frequency aliasing.

## 5. Discussions and Conclusions

In summary, we developed a chromatic LED array with multi-hyperuniform structure, that is, luminous points of each monochromatic chip exhibited hyperuniformity independently, and red-green-blue luminous points combined to show hyperuniformity as well. The chromatic LED array was developed to mimic the virtues of avian retina optical sampling, specifically, suppressing color misregistration and chromatic moiré fringes caused by periodic optical sampling. The placement and orientation of LED array has an impact on whether the arrangement of LED array is multi-hyperuniform, and two random procedures are used here to ensure LED array is of multi-hyperuniformity. Comparisons were performed numerically and experimentally by reconstructing images by different sampling methods in a single-pixel imaging system. Both numerical and experimental results indicated that the multi-hyperuniform sampling method yielded images with better image quality compared to the other two methods by using the same and basic interpolation algorithm. Besides the improvement of the images’ quality, the proposed imaging system achieved 208 fps frame rate experimentally, which has a potential in high dynamic applications.

This work is a proof-of-principle to demonstrate the feasibility of multi-hyperuniformity in high dynamic chromatic optical sampling. The chromatic LED array developed in this work contains only 1024 chromatic luminous LED chips due to the lab manufacturing limitation. The capability of improvement of image quality is almost the same by 32 × 32 multi-hyperuniform LED array with different orientations and placements, which can be enhanced by integrating more LED chips on the sampling array due to the property of the hyperuniform structure. Although the low-resolution color images are reconstructed, the method in this paper can offer a new solution to suppress the artificial effects in high resolution imaging which would not increase imaging time. Cutting-edge LED manufacturing techniques such as micro-LED or OLED could be used to develop multi-hyperuniform LED array with a much larger chip number and a higher density to take full advantages of such a sampling structure in high-resolution imaging. In this work, the multi-hyperuniform pattern has been verified to be able to improve image quality with a high frame rate. In future, LEDs could be used not only for illumination and display, but also for the development of new bionic imaging sensors.

## Figures and Tables

**Figure 1 sensors-21-04084-f001:**
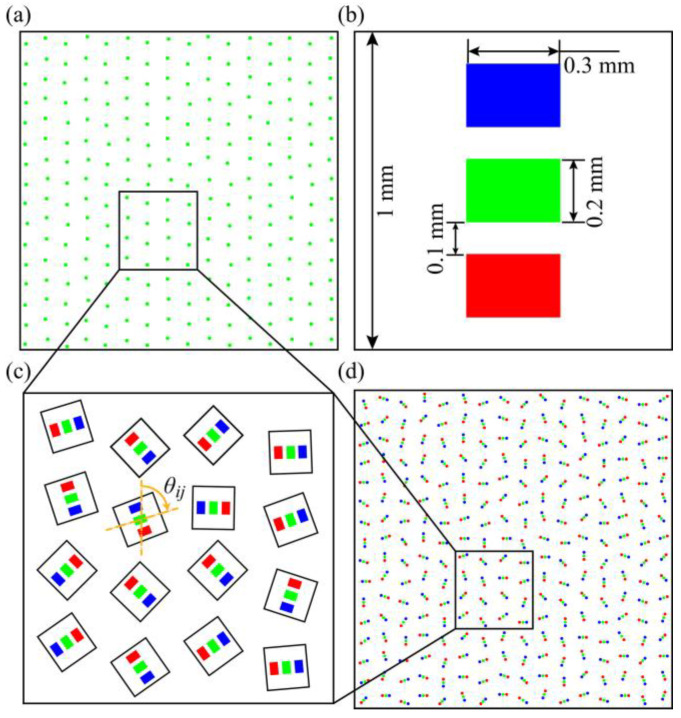
Schematics of multi-hyperuniform sampling arrangement: (**a**) Hyperuniform point pattern generated by the randomization procedure for green luminous point; (**b**) The geometry of a single LED chip; (**c**) Randomized rotations were introduced to each LED chip; (**d**) Multi-hyperuniform point pattern of the LED array.

**Figure 2 sensors-21-04084-f002:**
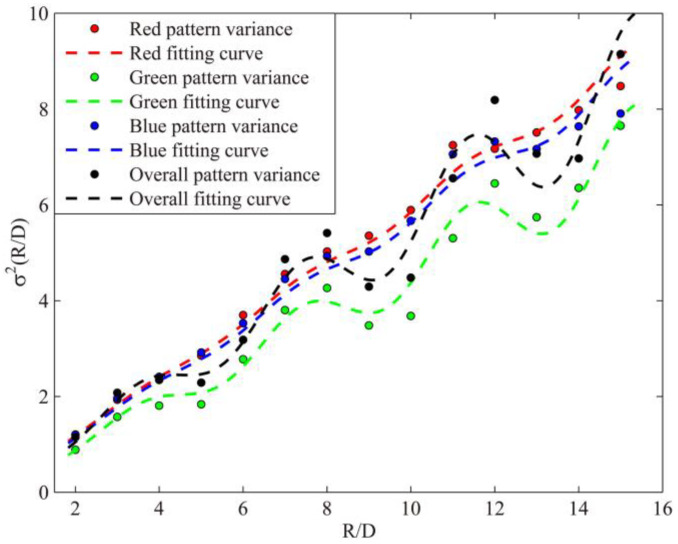
Number variance *σ*^2^*(R/D)* of all monochromatic point patterns and the overall point pattern, and their corresponding fitting curves.

**Figure 3 sensors-21-04084-f003:**
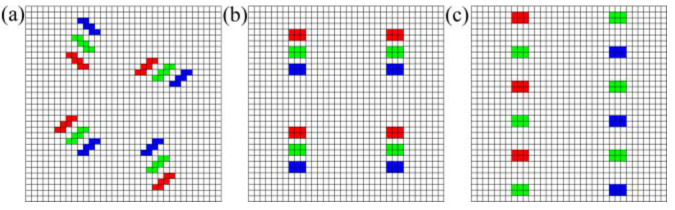
Partial areas of sampling patterns containing 12 sampling points: (**a**) Multi-hyperuniform sampling pattern; (**b**) Regular sampling pattern; (**c**) Bayer sampling pattern.

**Figure 4 sensors-21-04084-f004:**
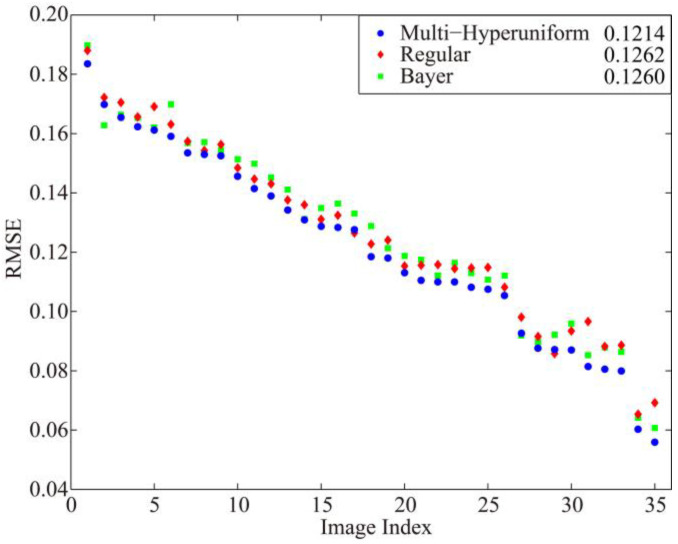
RMSEs of the reconstructed chromatic images using multi-hyperuniform (**blue dot**), regular (**red diamond**), and Bayer (**green square**) sampling patterns. The averaged values of 35 image RMSEs are listed on the label.

**Figure 5 sensors-21-04084-f005:**
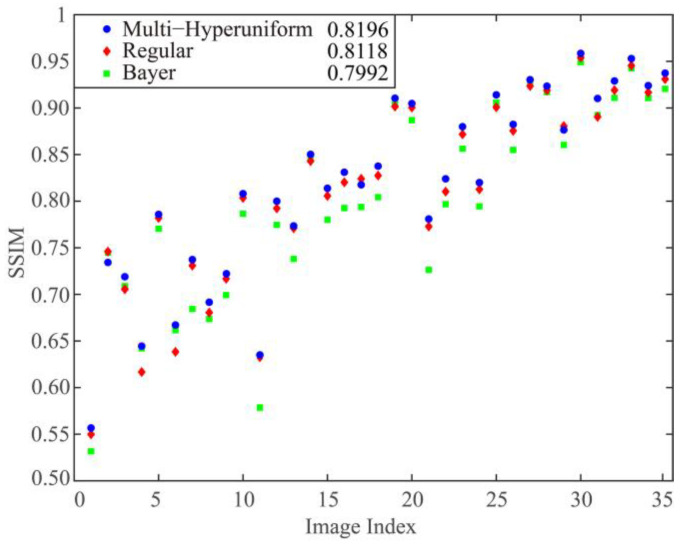
The SSIM values of the reconstructed chromatic images using multi-hyperuniform (**blue dot**), regular (**red diamond**), and Bayer (**green square**) sampling patterns. The averaged values of 35 image SSIMs are listed on the label.

**Figure 6 sensors-21-04084-f006:**
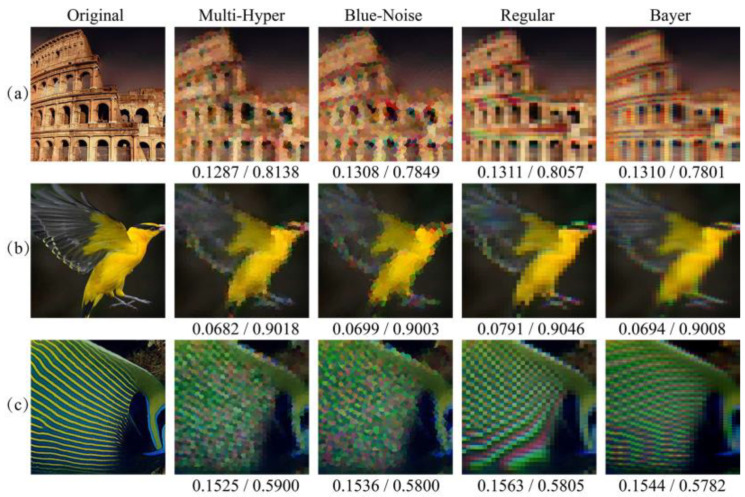
Comparison of reconstructed images of artificial scenes using different sampling patterns. From the left to right are the original images, reconstructions from multi-hyperuniform, regular and Bayer sampling patterns. The corresponding RMSEs and SSIMs are listed below the images, respectively. (**a**) The scene of Colosseum with color transition area and corresponding reconstructed images in numerical simulations; (**b**) The scene of bird with periodic patterns on the wings and corresponding reconstructed images; (**c**) The scene of fish with periodic stripes and corresponding reconstructed results.

**Figure 7 sensors-21-04084-f007:**
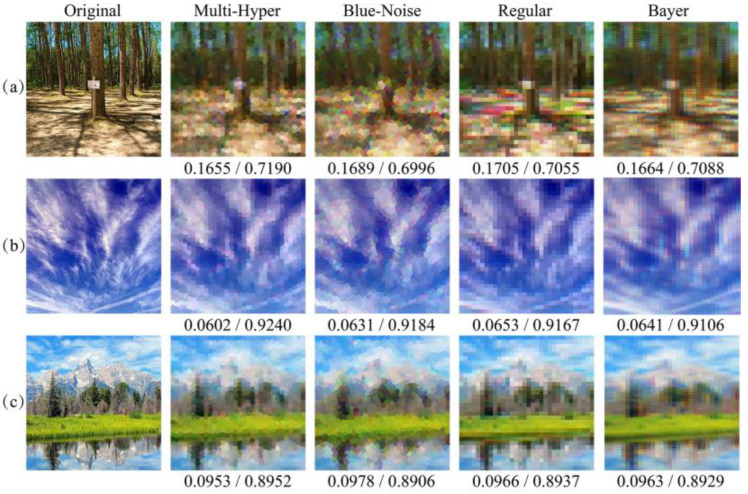
Comparison of reconstructed images of natural scenes using different sampling patterns. From the left to right are the original images, reconstructions from multi-hyperuniform, blue-noise sampling patterns, regular and Bayer sampling patterns. The corresponding RMSEs and SSIMs are listed below the images, respectively. (**a**) The scene of forest and corresponding reconstructed image; (**b**) The scene of sky and corresponding reconstructed images; (**c**) Natural scene with the different contents and corresponding reconstructed results.

**Figure 8 sensors-21-04084-f008:**
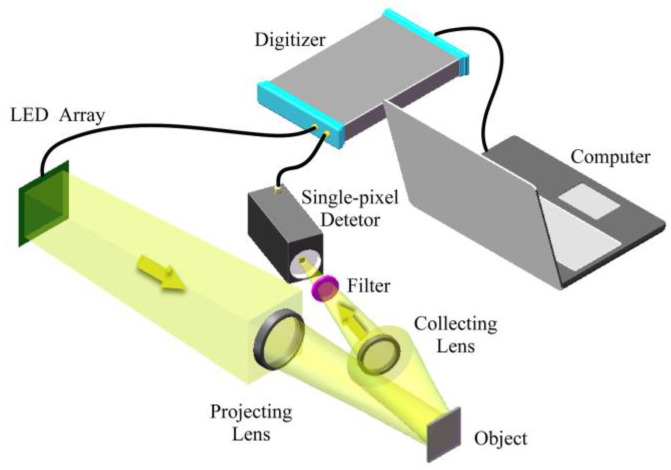
Experimental set-up. A multi-hyperuniform chromatic 32 × 32 × 3 LED array provided structured illumination to sample the object through a projection lens (f = 150 mm). A single-pixel bucket detector (Thorlabs PMT2102) and a digitizer (PicoScope 6404D) collected the reflected light intensity and transferred it to a computer for reconstruction.

**Figure 9 sensors-21-04084-f009:**
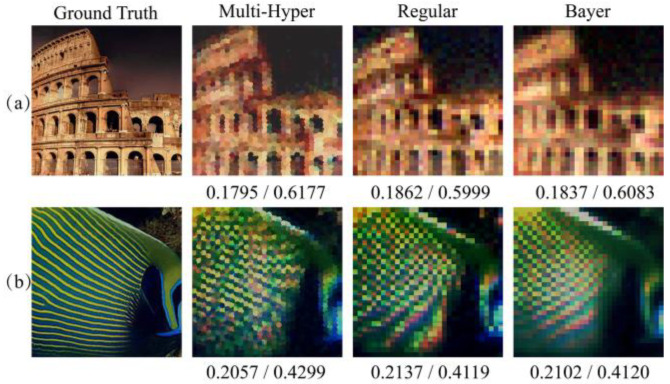
Experimental results of two group scenes. Chromatic images are reconstructed experimentally by multi-hyperuniform, regular and Bayer sampling patterns with their RMSEs and SSIMs listed below. (**a**) The scene of Colosseum with color transition area and corresponding reconstructed results in experiments; (**b**) The scene of fish with periodic stripes and corresponding recon-structed results.

**Table 1 sensors-21-04084-t001:** Fitting coefficients for different types of point patterns.

	Green	Red	Blue	Overall
P	0.4675	0.5932	0.5725	0.5663
Q	0.1274	0.0271	0.0335	0.1505

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
