# Peer review of "Bionic Birdlike Imaging Using a Multi-Hyperuniform LED Array"

_sensors, 2021, doi:10.3390/s21124084_

Round 1
Reviewer 1 Report
This paper designs a chromatic LED array in which red‐green-blue luminous points formed a multi‐hyperuniform arrangement to mimic the multi‐hyperuniform structure of the chicken retina system. Compared with periodic arrangement of Bayer array, it introduces less frequency aliasing in sampling and color misregistration. The optical sampling using multi‐hyperuniformity was performed experimentally via a single‐pixel imaging system.
I mainly concern about four issues:
1) The authors only compared the proposed random sampling strategy with regular structures. Since anlising errors are usually caused by insufficient sampling, it is necessary to compare the proposed sampling with other random sampling strategies, such as blue-noise sampling strategy.
2) Imaging quality is affected by both array structure and demosaicing method, however the used demosaicing method is highly related to the array structure. The authors should give more details of the demosaicing, and try at least two different state-of-the-arts demosaicing methods(e.g. compressive-sensing-based, deep-learning-based) to prove the advance of the proposed array.
3) The authors should also give more results of various scenes for comparisons. As we know, different images have different power distribution of image spatial frequency(please refer to(1)and(2)). I thought the proposed array would perfom differently on various scenes, the performance on natural scenes might be much better than that on artificial scenes. Also, the authors should cite the two related papers:
(1) Farinella, Giovanni Maria, et al. "Natural versus artificial scene classification by ordering discrete fourier power spectra." Joint IAPR International Workshops on Statistical Techniques in Pattern Recognition (SPR) and Structural and Syntactic Pattern Recognition (SSPR). Springer, Berlin, Heidelberg, 2008.
(2) Li, Yuqi, et al. "Optimized multi-spectral filter array based imaging of natural scenes." Sensors 18.4 (2018): 1172.
4) It is not clear why the designed R-G-B channels have the same spatial density since human eyes are most sensitive to green light.
Reviewer 2 Report
The submitted manuscript describes a chromatic LED array with multi-hyperuniformity structure. The main advantage of the described LED array is that can suppress color misregistration and chromatic moiré fringes caused by periodic optical sampling. The authors demonstrated the feasibility of multi-hyperuniform in high dynamic chromatic optical sampling. Meanwhile, the proposed method provides a new idea for the further studies on bionic imaging sensors.
Overall, the paper is very interesting, and the LED array proposed has great potential applications. The principle is explained clearly, the results are impressive. I have some questions for the authors to clarify.
1. In the section 3, “the gradient-based interpolation algorithm was applied in this work”, what’s the advantages of the chosen algorithm over other algorithms?
2. How to ensure that the demosaicking algorithm for the multi-hyperuniform sampling pattern is also the best algorithm for other sampling patterns?
3. There are some mistakes in the description. For example, in Equation (2) and Table 1, what does P mean? On line 178 of page 5, "demosacing" should be "demosaicking".
Reviewer 3 Report
Kindly see the attached comments

Round 2
Reviewer 1 Report
The authors gave reasonable explanations and analyses of the issues I raised. The current version of the paper is appropriated to be published on Sensors.